# A Review on Emerging Pollutants in the Water Environment: Existences, Health Effects and Treatment Processes

**Nor Zaiha Arman [1], Salmiati Salmiati [1,2], Azmi Aris [1,2], Mohd Razman Salim [3], Tasnia Hassan Nazifa [4], Mimi Suliza Muhamad [5] and Marpongahtun Marpongahtun [6,***

1 Center for Environmental Sustainability and Water Security (IPASA), Research Institute for Sustainable Environment (RISE), Universiti Teknologi Malaysia, Skudai 81310, Malaysia; n.zaiha@utm.my (N.Z.A.); salmiati@utm.my (S.S.); azmi.aris@utm.my (A.A.)
2 Department of Environmental Engineering, School of Civil Engineering, Faculty of Engineering, Universiti Teknologi Malaysia, Bahru 81310, Malaysia
3 Civil Engineering Department, Faculty of Engineering, Technology and Built Environment, UCSI University, Cheras, Kuala Lumpur 56000, Malaysia; razman@ucsiuniversity.edu.my
4 School of Engineering and Applied Sciences, Memorial University of Newfoundland, St. John's, NL A1C 5S7, Canada; thnazifa@mun.ca
5 Department of Civil Engineering Technology, Faculty of Civil Engineering Technology, Universiti Tun Hussein Onn Malaysia, Pagoh Education Hub, Batu Pahat 84600, Malaysia; msuliza@uthm.edu.my
6 Department of Chemistry, Faculty of Mathematics and Natural Science, Universitas Sumatera Utara, Medan 20155, Indonesia
* Correspondence: marpongahtun@usu.ac.id

**Abstract:** Emerging pollutants (EPs), also known as micropollutants, have been a major issue for the global population in recent years as a result of the potential threats they bring to the environment and human health. Pharmaceuticals and personal care products (PPCPs), antibiotics, and hormones that are used in great demand for health and cosmetic purposes have rapidly culminated in the emergence of environmental pollutants. EPs impact the environment in a variety of ways. EPs originate from animal or human sources, either directly discharged into waterbodies or slowly leached via soils. As a result, water quality will deteriorate, drinking water sources will be contaminated, and health issues will arise. Since drinking water treatment plants rely on water resources, the prevalence of this contamination in aquatic environments, particularly surface water, is a severe problem. The review looks into several related issues on EPs in water environment, including methods in removing EPs. Despite its benefits and downsides, the EPs treatment processes comprise several approaches such as physico-chemical, biological, and advanced oxidation processes. Nonetheless, one of the membrane-based filtration methods, ultrafiltration, is considered as one of the technologies that promises the best micropollutant removal in water. With interesting properties including a moderate operating manner and great selectivity, this treatment approach is more popular than conventional ones. This study presents a comprehensive summary of EP's existence in the environment, its toxicological consequences on health, and potential removal and treatment strategies.

**Keywords:** water environment; emerging pollutants (EPs); treatment processes; health effects; removal strategies

## 1. Introduction

EPs have become a significant concern for the global population in recent years, owing to the possible dangers posed to the environment and human health. EPs, also known as micropollutants, are produced by various sources, including synthetic and natural substances. A new class of chemical compounds known as new EPs have recently been discovered in surface water, food sources, municipal wastewater, groundwater, and even drinking water. Such pollutants, known as EPs, are chemical composites that are commonly found in the environment, particularly in soil and aquatic bodies. However, they have

only recently been identified as significant water contaminants. Personal care products (PCPs), hormones, flame retardants, industrial additives, endocrine-disrupting chemicals (EDCs), pharmaceuticals, nanomaterials, and pesticides are examples of EPs that are widely used and indispensable in modern society [1–4]. Based on the NORMAN network, at least 700 substances classified into 20 classes were identified in the European aquatic environment [5]. US Geological Survey characterized EP as "any compound of engineered or normal root or any microorganism that is not usually observed in the surrounding, however it can possibly cause unfriendly environmental and additionally human wellbeing impacts". These contaminants are usually found in trace concentration from few parts per trillion to parts per billion [6]. According to Dulio et al. [7], the term "EPs" are substances that can persist in the environment, bioaccumulate, and potentially life-threatening, such as causing abnormal growth, reduced fertility and reproductive health, neurodevelopmental delays, inhibiting wildlife species, degrading aquatic ecosystems, and possibly harming the human immune system [8]. It is important to highlight that the majority of emerging contaminants are not new or recently introduced pollutants into the environment. Most emerging contaminants, on the other hand, are well-established pollutants with a newly documented harmful effect or mode of action. Therefore, the term "emerging" applies to both the contaminant and the issues that have arisen. Hence, emerging contaminants are also known as "contaminants of emerging concern" or "chemicals of emerging concern". In a broader sense, emerging pollutants can be classified according to the following criteria: (i) not necessarily a new compound, (ii) a compound that has long existed in the environment but whose presence has only recently been detected and whose significance is beginning to be recognized, and (iii) a long-known compound whose potential negative impact on humans and the environment has only recently been realized. Table 1 lists the EP groups and their major compounds.

**Table 1.** Emerging pollutants groups and their major compounds [9–11] (Reprinted by permission).

| Groups/Examples | | Compounds |
|---|---|---|
| Pharmaceuticals | *Human antibiotics and veterinary* | Trimethoprim, erytromycine, amoxicillin, lincomycin, sulfamethaxozole, chloramphenicol |
| | *Analgesics, anti-inflammatory drugs* | Ibuprofene, diclofenac, paracetamol, codein, acetaminophen, acetylsalicilyc acid, fenoprofen |
| | *Psychiatric drugs* | Diazepam, carbamazepine, primidone, salbutamol |
| | *β-blockers* | Metoprolol, propanolol, timolol, atenolol, sotalol |
| | *Lipid regulators* | Bezafibrate, clofibric acid, fenofibric acid, etofibrate, gemfibrozil |
| | *X-ray contrasts* | Iopromide, iopamidol, diatrizoate |
| Personal care products | *Fragrances* | Nitro, polycyclic and macrocyclic musks, phthalates |
| | *Sun-screen agents* | Benzophenone, methylbenzylidene camphor |
| | *Insect repellents* | N,N-diethyltoluamide |
| Endocrine Disrupting Chemicals (EDCs) | | 4-octylphenol, cholesterol, estrone, 17β-estradiol, 17α-ethinylestradiol, coprostanol, progesterone, stigmasterol, 4-nonylphenol, Di(2-ethylhexyl) phthalate (DEHP), Bisphenol A (BPA) |
| Hormones and steroids | | Estradiol, estrone, estriol, diethylstilbestrol (DES) |
| Perfluoronated compounds (PFCs) | | Perfluorotoctane sulfonates (PFOs), perfluoroctanoic acid (PFOA) |
| Surfactants and surfactant metabolites | | Alkylphenol ethoxylates, 4-nonylphnol, 4-octylphenol, alkylphenol carboxylates |

**Table 1.** *Cont.*

| Groups/Examples | Compounds |
|---|---|
| Flame retardants | Polybrominated diphenyl ethers (PBDEs): polybromonated biphenyls (PBBs) – polybromonated dibenzo-*p*-dioxins (PBDDs) –polybromonated dibenzofurans (PBDFs), Tetrabromo bisphenol A, C10-C13 chloroalkanes, Tris (2-chloroethyl) phosphate, Hexabromocyclododecanes (HBCDs), Hydrophobic Brominated Compounds |
| Plasticizers | Di-2-propylheptyl phthalate (DPHP), Di-2-ethylhexyl terephthalate (DEHTP), Di-n-butyl adipate (DnBA), Di-isobutyl adipate (DIBA), Di-iso-nonyl adipate (DINA) |
| Industrial additives and agents | Chelating agents (EDTA), aromatic sulfonates |
| Gasoline additives | Dialkyl ethers, Methyl-t-butyl ether (MTBE) |
| Antiseptics | Triclosan, chlorophene, esters of p-hydroxybenzoic acid (parabens) |

Water resources quality has deteriorated due to contamination caused by urbanization, rapid population growth, agricultural activities, and industrial development [12]. Heavy metals, microbial pollutants, priority contaminants, and nutrients are the most commonly studied aspects of water quality. Nonetheless, recent research [11,13] revealed the presence of organic pollutants that have a significant impact on water parameters. The main issue with EPs is a lack of understanding about their long-term effects on aquatic life, the environment, and human health. The discovery of numerous new compounds in drinking, ground, and surface water has alarmed the public, mainly when human health-based guidelines are unavailable [14,15]. Numerous studies were carried out to determine the contaminants concentrations and sources in receiving water bodies [16,17]. Due to the various concentrations and the lack of systematic monitoring programs, information about their transformation products, metabolites, and drinking water treatment is still limited. Also, since most EPs are not subject to water and wastewater regulations, there is little information or data involved in water resources.

Nevertheless, policymakers have recently agreed that EPs must be addressed systematically and coherently, although many remain unregulated. The European Union (EU), for example, has established a complex set of regulatory frameworks for EPs for governing activities involving the commercialization, use, presence, and emissions of chemical pollutants in the environment. Additionally, through their relevant agencies, the United States (US) excelled in continuous monitoring practices and maximum limit regulations [18].

Sutherland and Ralph [19] presented an extensive review on the microalgal bioremediation potentials of emerging pollutants. They demonstrated that microalgae have ability to concentrate, filter, eliminate or biotransform a wide range of emerging pollutants. Gogoi et al. [20] investigated the fate and occurrence of these contaminants in wastewater treatment plants and in the environment. The writers indicated out that future research should concentrate on the improvement of risk-based screening framework and models. [8] carefully reviewed some treatment technologies such as biological, phase-change, advance oxidation process for removal of emerging pollutants from water. In addition, microplastics are classified as emerging pollutants, the interaction between pollutants and microplastics was evaluated by Abaroa-Pérez et al. [21]. Khan et al. [22] considered pharmaceuticals as emerging major source of pollution for the environment. They considered effluent discharge from hospitals and emphasized the treatment processes such as activate sludge process, sequencing batch reactor, membrane biological reactor, activated carbon treatment, carbon nanotubes treatment, upflow anaerobic sludge blank, UV/$H_2O_2$, Fenton, and ozone treatment. A recent study by Roy et al. [23] considered antibiotics as only form of

emerging water pollutant and focused on the treatment technologies such as photocatalytic degradation and in combination with nanomaterials. Monitoring, challenges of emerging pollutants, implementing efficient and ecological methods for their removal are well described by Vasilachi et al. [11].

Although huge scientific studies are available on several aspects concerning emerging pollutant's monitoring, analysis, we consider that it is urgently needed to review these research efforts in a holistic way from occurrence, distribution, categorize to treatment of such emerging pollutants. To our best knowledge, there is not yet a review paper that structures all the research efforts related to the occurrence, effects of emerging pollutants in water environment and different technological options. As a result, this review script aims to bring comprehensive literature studies till date regarding the occurrences of emerging pollutants in water sources, distribution, analytical methods, toxicological effects and treatment processes.

This review paper gave insight into EPs in the water environment, which recently received much attention. In this paper, peer-reviewed scientific literature on emerging pollutants or emerging contaminants were reviewed with special regards to their occurrence, detection techniques, analytical methods, fate in the environment, and toxicity assessment. The discussion in this research also involved the current state of various water treatment processes for EP removal. This review demonstrated that EPs might pose a significant risk to consumers. However, there is substantially limited information on the by-products' formation and their toxicity. Therefore, more research is required to better understand the EPs that exist in the water environment and elucidate the entire degradation pathway. A total of 4000 documents (among research articles and review papers) from Scopus databases were appeared while using keywords occurrence, detection techniques, analytical methods, fate in the environment, and toxicity assessment. The current study is prepared by reviewing 238 scientific articles, with 185 review publications and the rest are research publications. These studies are from different countries across the world, published between the 2011 to 2021.

## 2. Sources of EPs and Their Occurrence in Water Resources

Both surface and groundwaters have been found to contain EPs. Due to dilution and natural attenuation processes, their concentrations in surface waters are frequently lower than those recorded immediately at the outflow of wastewater and sewage treatment plants. Surface water, on the other hand, has higher EPs concentrations than groundwater since it receives effluent directly from the WWTP and has a shorter residence time. But, if the aquifer is close to pollution sources, groundwater concentrations could rise. A few researchers have discovered that the occurrence of certain PCPs and pharmaceuticals in surface water varies considerably. The variability is highly probable due to the usage frequency and dosages in different regions, including the effectiveness of the WWTP system [24]. Numerous factors influence the transport, fate, and occurrence of EPs in the surroundings. Among them are the physico-chemical properties of the environment and water, as well as longitude and latitude. Furthermore, the source type influences the exposure degree and the substance's properties [9]. Figure 1 depicts the sources and possible EPs routes released into the atmosphere and dissolving into various receptors (ground, surface, and drinking water).

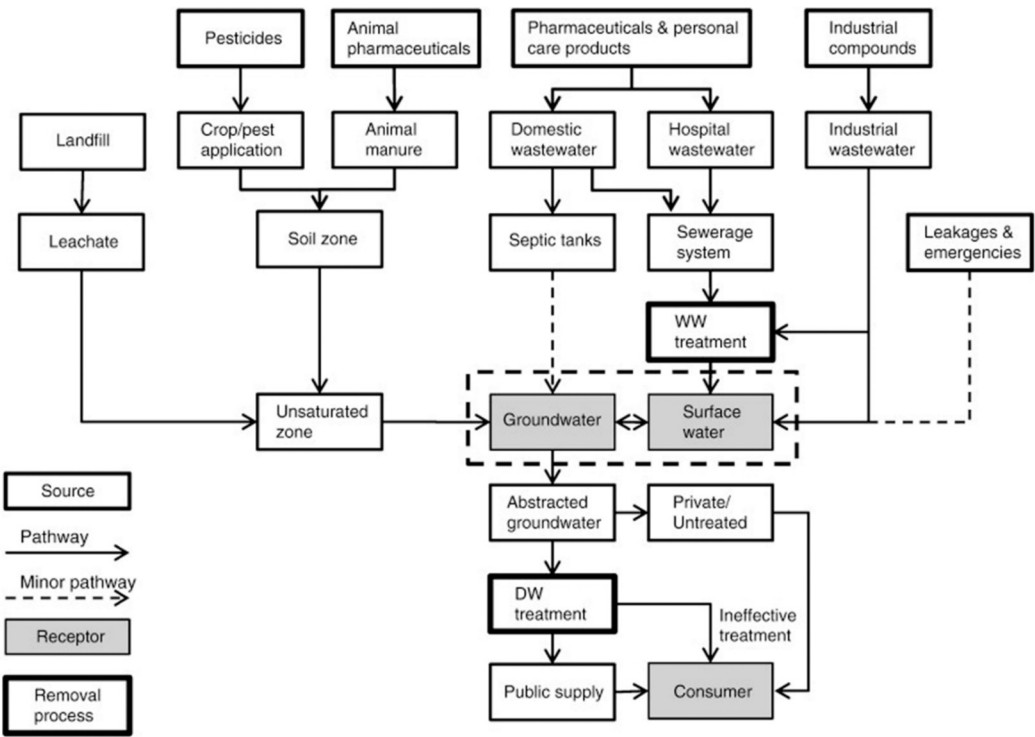

**Figure 1.** Sources and potential routes of EPs in the water environment [25]. (Reproduced from [25] with Elsevier permission, license 5191080813331, from 16 November 2021).

### 2.1. Pharmaceuticals and Personal Care Products (PPCPs)

PCPs include various chemicals, including cosmetics and health care products (e.g., over-the-counter drugs, supplements, and prescription pharmaceuticals). After being released from leached farmland manure or sewage treatment plants, these pollutants end up in the community water systems and soil [26]. The pharmaceutical sector has a massive presence in WWTP operations [27]. Based on research made in the Ter River, Catalonia, Spain, the effect of effluent from WWTP is minimal compared to the river's widespread presence of compounds. Likewise, research in groundwater on Cape Cod, Massachusetts, found that sewer systems from residential and industrial development, on top of other sources such as on-site wastewater treatment for nursing homes and health services, are a significant source on the incidence of EPs [28].

### 2.2. Antiseptics

On the other hand, triclosan is an antiseptic/antibacterial agent frequently found in household appliances, toothpaste, playthings, soaps, clothing, bedding, plastic, and fabric. Triclosan in tap water produces chloroform as one of the chlorinated by-products. Triclosan can break down in the environment to methyl triclosan or certain dioxins, particularly 2,4-dichlorophenol (2,4-DCP) and 2,8-dichlorodibenzo-p-dioxin (2,8-DCDD) [29]. The United States Geological Survey (USGS) of 95 different organic wastewater pollutants in the US streams revealed that at the highest concentrations, triclosan was one of the frequently detected compounds. Triclosan has also been discovered in numerous water bodies throughout the US, and researchers suspected that conventional treatment methods have not entirely removed it [30]. Additionally, the safety standards of triclosan in drinking water have yet to be established.

### 2.3. Hormones and Steroids

Among the various nonprescription and prescription or over-the-counter drugs are synthetic and natural steroid hormones such as gestagens, estrogens, and androgens [31].

These drugs are used for various purposes in humans and animals since they can modify their physiology and trigger essential regulatory functions in the body fluid. Ferguson et al. [32] discovered that in South-eastern Australia, the estrogens entering the estuary are most likely coming from a nearby WWTP. Its detection in freshwater indicates a secondary source, most likely from the agricultural area. Another study discovered that crop residues were the primary source of steroid estrogens, with the conjugated estrogens accounting for up to 22% of the total estrogen load from dairy farming. Furthermore, estrogens have been found in soil drainage water, streams draining stock grazing fields, runoff from dairy wastes application, on groundwater under unlined effluent holding ponds, and land [33].

### 2.4. Perfluoronated Compounds (PFCs)

PFCs are a type of compound that is applied in paints, food packaging, textiles, adhesives, polishes, waxes, electronics, and stain repellents, among other things. The most common are perfluorocarbon sulfonic acids (PFSAs) and perfluorinated carboxylic acids (PFCAs). In contrast, perfluoro octane sulfonate (PFOS) and perfluorooctanoic acid (PFOA) are the most commonly known usage [34]. Due to releases from non-point sources, WWTPs, and industrial facilities, PFOA is detected in drinking and treated water sources [35]. It could also be present in drinking water wells via the contaminated groundwater plume passage.

Additionally, PFOA can enter groundwater through the air from nearby industrial sites before deposition onto soil and seepage into groundwater [36,37]. Pitter et al. [38] and Steenlan et al. [39] discovered that the release of PFOA in Ohio and West Virginia from an industrial facility could contaminate drinking water wells up to 20 miles away. This situation happened when PFOA-containing air pollution from an industrial source settled in the ground, accompanied by passage to groundwater and recharge the groundwater aquifer with contaminated surface water from the Ohio River.

### 2.5. Disinfection By-Products (DBPs)

On the other hand, the primary source of disinfection by-products (DBPs) came from the drinking water treatment plant (WTP) [40–42]. DBPs were determined in 15 WTPs in Beijing City from various water sources by Stalter et al. [40]. They discovered that halogenic acetic acids (HAAs) and trihalomethane (THMs) accounted for 38.1% and 42.6% of all DBPs, respectively, in all treated samples. DBPs in drinking water were found to vary with a water source, with surface water having the highest levels, mixed water sources, and groundwater having the lowest levels.

### 2.6. Pesticides

Pesticides can contaminate drinking water due to carelessness, such as back-siphoning, application in lawns and golf courses, and a sizeable accidental spill [43]. Pesticide metabolites have been highly prone to leaching in soil [44,45]. Reemtsma et al. [46] discovered pesticide metabolites were found in ground and surface water, with the diversity in the runoff samples attributed to multiple pesticide applications in minimal urban areas. Various metabolites in water matrices indicate that pesticides are widely used in city centers [46,47]. Belenguer et al. [48] discovered pesticides concentration and presence such as prochloraz, clofenvinphos, pyriproxyfen, imazalil, and dichlofenthion, in water are linked to intensive agricultural activities in the area. Table 2 shows a compilation of studies on EPs concentrations in various water bodies around the world.

**Table 2.** Concentrations of EPs in water samples worldwide.

| Country | Sample | Compound | Concentration (ng/L) | Reference |
|---|---|---|---|---|
| Kenya | Groundwater | Paracetamol<br>Metronidazole<br>Carbamazepine | 10–30<br>7–10<br>30–40 | [49] |
| Japan | Drinking water | Acemetacin<br>Acetaminophen<br>Antipyrine<br>Aspirin<br>Diclofenac<br>Diflunisal | 5<br>2.8<br>8.3<br>6<br>2.5<br>2.1 | [50] |
| China | Raw water | Acetaminophen<br>Antipyrine<br>Carbamazepine | 15.2<br>3.8<br>0.8 | [4] |
| | Surface water | DEET<br>Carbamazepine | 0.8–10.2<br>0.01–3.5 | [51] |
| | Surface water | E1<br>E2<br>EE2<br>E3 | 22.7<br>6.5<br>4.4<br>5.3 | [52] |
| | Surface water | E1<br>E2<br>EE2<br>E3<br>DES<br>EV | 49.8<br>11.5<br>24.4<br>14.2<br>2.12<br>7.66 | [53] |
| | Surface water | E1<br>E2<br>EE2<br>E3<br>DES<br>EV | 2.98<br>1.78<br>2.67<br>4.37<br>2.52<br>1.96 | [54] |
| | Surface water | E1<br>E2<br>EE2<br>E3 | 14–180<br>n.d–134<br>7–24<br>4.94 | [55] |
| | Surface water | E1<br>E2<br>E3 | n.d–3.80<br>n.d–0.97<br>n.d–5.79 | [56] |
| | Surface water | PFOS<br>PFOA<br>PFHpA<br>PFNA<br>PFDA | 0.5<br>0.1<br>0.5<br>0.1<br>0.5 | [57] |
| Netherlands | Drinking water | Oxazepam<br>Temazepam<br>Benzoylecgonine | 3–13<br>1–10<br>1–3 | [58] |

**Table 2.** *Cont.*

| Country | Sample | Compound | Concentration (ng/L) | Reference |
|---|---|---|---|---|
| Spain | Groundwater | Cocaine | 60.2 | [59] |
| | | Benzoylecgonine | 19.6 | |
| | | Cocaethylene | 1.8 | |
| | | Morphine | 27.2 | |
| | | Methadone | 68.3 | |
| | | EDDP | 8.2 | |
| | Drinking water | MDMA | 36.8 | [60] |
| | | MDMA | 1.51 | |
| | | Benzoilecgonine | 2.47 | |
| | | Cocaine | 2.11 | |
| | | Methadone | 0.47 | |
| | | EDDP | 0.34 | |
| | | Ephedrine | 0.27 | |
| | Drinking water | Caffeine | 392 | [61] |
| | | Nicotine | 141 | |
| | | Cotinine | 9.8 | |
| | | Cocaine | 2.3 | |
| | | Cocaethylene | 0.9 | |
| | | Benzoylecgonine | 3.1 | |
| | | AMP | 1.7 | |
| | | MDA | 0.9 | |
| | | METH | 1.4 | |
| | | MDEA | 0.6 | |
| | Surface water | 1H-Benzotriazole | 16 | [62] |
| | | Nonyphenol Monoethoxylate | 0.4 | |
| | | Nonyphenol | 24 | |
| | | Octyphenol | 6.8 | |
| | | Bisphenol A | 27.6 | |
| | | E1 | 17 | |
| | | E2 | <0.037 | |
| | | EE2 | <0.14 | |
| | | E3 | <0.17 | |
| | | DES | <0.043 | |
| | Surface water | E1 | 1 | [63] |
| | | EE2 | 3.4 | |
| | | E3 | 72 | |
| | Surface water | PFOA | 14–22.4 | [64] |
| | | PFHpA | 7–14.3 | |
| | | PFNA | 5–33.7 | |
| | | PFDA | 0.5–36.7 | |
| | Surface water | PFOS | 0.01–42.6 | [65] |
| | | PFOA | 2–188.6 | |
| | | PFHpA | 0.4 | |
| | | PFNA | 87.4 | |
| | | PFDA | 0.1–13 | |

**Table 2.** *Cont.*

| Country | Sample | Compound | Concentration (ng/L) | Reference |
|---|---|---|---|---|
| Canada | Drinking water | Carbamazepine | 601 | [66] |
| | | Erythromycin | 155 | |
| | | Ibuprofen | 25 | |
| | | Lincomycin | 1413 | |
| | | Gemfibrozil | 4 | |
| | | Monensin Na | 76 | |
| | | Tylosin | 31 | |
| | | Tetracycline | 15 | |
| | | Enrofloxacin | 13 | |
| | | Roxithromycin | 41 | |
| | | Bezafibrate | 1 | |
| | | Sulfamethoxazole | 2 | |
| | | Acetaminophen | 17 | |
| | | Trimethoprim | 15 | |
| France | Drinking water | Carbamazepine | 41.6 | [67] |
| | | Oxazepam | 57 | |
| | | Paracetamol | 71 | |
| | | Atenolol | 34 | |
| | | Bezafibrate | 12.4 | |
| | | Diclorafenac | 35 | |
| | | Fenofibric acid | 1 | |
| | | Ibuprofen | 8 | |
| | | Ketoprofen | 22 | |
| | | Lorazepam | 0.7 | |
| | | Metoprolol | 2.0 | |
| | | Metronidazole | 0.1 | |
| | | Naproxen | 6.4 | |
| | | Pravastatine | 1.6 | |
| | | Propranolol | 2.0 | |
| | | Roxithromycine | 18.1 | |
| | | Salicylic acid | 29.0 | |
| | | Sulfamethoxazole | 4.0 | |
| | | Trimethoprime | 2.0 | |
| United States | Groundwater | Acetaminophen | 1.89 | [68] |
| | | Caffeine | 0.29 | |
| | | Carbamazepine | 0.42 | |
| | | Codeine | 0.214 | |
| | | p-Xanthine | 0.12 | |
| | | Sulfamethoxazole | 017 | |
| | Drinking water | Erythromycin-$H_2O$ | 1.5 | [69] |
| | Groundwater | Trimethoprim | 1.5 | [28] |
| | | Sulfamethoxazole | 113 | |
| | | Phenytoin | 66 | |
| | Surface water | E1 | 6 | [70] |
| | | E2 | 2 | |
| Brazil | Surface water | E1 | <16 | [71] |
| | | E2 | 6806 | |
| | | EE2 | 4390 | |
| Korea | Surface water | PFOS | 4.11–450 | [72] |
| | | PFOA | 2.95–68.6 | |
| | | PFNA | 1.38–14.7 | |
| | | PFDA | 0.23–15.4 | |

**Table 2.** *Cont.*

| Country | Sample | Compound | Concentration (ng/L) | Reference |
|---|---|---|---|---|
| Europe (Germany, Austria, Slovakia, Hungary, Croatia, Serbia, Romania, Bulgaria, Moldova and Ukraine) | Surface water | PFOS | 0.08–19 | [73] |
| | | PFOA | 0.1–46 | |
| | | PFHpA | 0.2–3 | |
| | | PFNA | 0.05–2 | |
| India | Surface water Groundwater | PFOS PFOS | 0.025 0.033 | [74] |
| | Surface water | PFOA PFOS | 4–93 3–29 | [75] |
| | Surface water | PCBs | 16.1–23.3 | [76] |
| Malaysia | Surface water | propiconazole pymetrozine Imidacloprid Tebuconazole | 17.6–4493.1 1.3–260.8 4.6–57.7 4.0–512.1 | [77] |
| | Drinking water | Ethinylestradiol Norgestrel Metoprolol | 130 30 39 | [78] |

MDA: 3,4-Methylenedioxyamphetamine; MDMA: 3,4-Methylenedioxymethamphetamine; MDEA: N-Methyldiethanolamine; AMP: Alpha-amino-3-hydroxy-5-methyl-4-isoxazolepropionic acid; METH: Methamphetamine; DEET: N,N-Diethyl-meta-toluamide; EDDP: 2-Ethylidene-1,5-dimethyl-3,3-diphenylpyrrolidine; E1: Estrone; E2: 17β-Estradiol; EE2: 17α-Ethinylestradiol; E3: Estriol; DES: Diethylstilbestrol; EV: Estradiol valerat; PFOA: Perfluorooctanoic acid; PFOS: Perfluorooctane sulfonate; PFHpA: Perfluoroheptanoic acid; PFNA: Perfluorononanoic acid; PFDA: Perfluorodecanoic acid; TCM: Chloroform; BDCM: Bromodi-chloromethane; DBCM: Dibromochloromethane; TBM: Bromoform; PCBs: Polychlorinated bi-phenyls.

## 3. Toxicological Effects of EPs

EPs' adverse effects have become a significant source of concern in society and the environment since they can cause cancer and endocrine disruption. EDCs are synthetic or natural chemicals that can block or mimic hormones and affect the living organisms' endocrine systems [79]. This disruption could affect normal hormone levels, stimulating or inhibiting hormone production and metabolism [80,81]. EDCs are EP classes that include phthalates, polybrominated compounds, polychlorinated biphenyls (PCBs), steroid sex hormones, pesticides, pharmaceutical products, bisphenol A (BPA), alkylphenol ethoxylates, and alkylphenols. When a mother is exposed to EDCs such as BPA and phthalates, the sexual development of her offspring can be hampered [82–84]. Diethyltoluamide/insect repellents (DEET), ultraviolet (UV) screens, synthetic musk fragrances, and parabens are types of PCPs that may also act as EDC in water [81,85–89].

### 3.1. Hormones and Steroids

Several hormones, such as estrone (E1), 17-oestradiol (E2), 17-ethinylestradiol (EE2), estriol (E3), equilin, 17-oestradiol, norethindrone, equilenin, and mestranol, are listed as priority drinking water contaminants based on their health effects and environmental occurrence [90]. At low concentrations, 17-ethinylestradiol can cause estrogenic effects in fish. The impacts also include changes in sexual characteristics and sex ratios that cause decreasing egg fertilization in fish [90] and feminization in fish [91,92]. Androgen hormones also influence fish masculinization, whereas glucocorticoids impair reproduction and immune system development [93]. Endocrine disruptors, such as phytoestrogens,

could also cause teratogenic, estrogenic, and other physiological problems in mammalian embryos and fish [94,95].

### 3.2. Antiseptics

Beauty and skincare products containing triclocarban and triclosan can cause the body's hormonal or endocrine disruptions [96]. Furthermore, these ingredients are harmful to fish embryos [97], as well as algae, crustaceans, and fish [98,99]. Triclosan, as per [100], can disrupt the reproductive axis and thyroid hormone homeostasis. Triclosan also has been shown to harm phytoplankton accumulation in freshwaters at specific concentrations in the environment. Long-term exposure to parabens, even at low concentrations, might cause vitellogenin synthesis in fish [101].

### 3.3. Plasticizers

Since BPA is among the most commonly used industrial added chemicals, its adverse health effects have been extensively researched. Rochester et al. [102] and Weber et al. [103] conducted a comprehensive review of the impact of BPA on human health and wildlife. BPA has been linked to lower sperm quality, fertility, sex hormone concentrations, and self-reported sexual function in men. In women, exposure is linked to polycystic ovary syndrome, breast cancer, miscarriage, endometrial disorders, and premature births.

### 3.4. Flame Retardants

In breast milk, the levels of polybrominated diphenyl ethers (PBDEs) are linked to lower body mass and birth weights, as well as cryptorchidism (undescended testicles) in newborn boys [104,105]. Dorman et al. [106] discovered PBDE levels in humans and animals that cause neurotoxicity development in animal model experiments, implying that PBDEs may have a similar effect on humans. Due to their lipophilicity, PBDEs are known to bioaccumulate in fat tissue. They have also been linked to harmful animal health risks such as fetal malformations, hormone disruption, and decreased sperm count [29].

### 3.5. Disinfection By-Products (DBPs)

Congenital disabilities, early-term miscarriage, and bladder cancer have been associated with DBPs in drinking water. Meanwhile, DBPs in swimming pools can cause respiratory problems or asthma [107]. The cancer risk from inhalation, ingestion, and dermal contact exposure to THMs was evaluated by Gan et al. [108]. They discovered that bromodichloromethane ($CHCl_2Br$) led to the most cancer risk from the ingestion pathway, while chloroform ($CHCl_3$) contributed the most cancer risk from the inhalation pathway.

### 3.6. Pesticides

Pesticides include dichlorodiphenyltrichloroethane (DDT), chlordane, vinclozolin, endosulfan and dieldrin, aldrin, and atrazine. Those components can disrupt the endocrine system by causing hormonal imbalances [109]. Moreover, organochlorine accumulation was linked to a higher risk of several types of genotoxicity, human cancer, and mental and psychomotor development [110]. Nanomaterials have been shown in toxicological studies to be neurotoxic, cytotoxic, genotoxic, bactericidal, and ecotoxic. In rodents and humans, anatase ($TiO_2$), zinc oxide (ZnO), and nano-sized silicon dioxide ($SiO_2$) could cause pulmonary inflammation [111]. Moreover, once released into the environment, Argentum, a silver nanoparticle, could disrupt ecological equilibrium [112]. Table 3 shows the additional harmful effects of EPs on the environment and their exposure limits.

**Table 3.** Toxicological effects of emerging pollutants.

| Compound | Level of Exposure | Adverse Effects | Assays | Reference |
|---|---|---|---|---|
| Bisphenol A (BPA) | 250 µg/kg | Inhibited fatty acid uptake and oxidative decomposition in male mice liver | Vertebrate and invertebrate animals | [113] |
| | 0.23 ppt | Disruption of cell function | Rat | [114] |
| | $10^{-12}$ M or 0.23 ppt | Stimulate calcium influx and prolactin secretion in rat pituitary tumor cells | Human urine | [115] |
| Phthalates | 0.3–345 µg/g creatinine | Toxic effects in the reproductive system | Sewage treatment plant effluents; surface; ground and drinking water | [116] |
| Triclosan | 1.4–3000 µ/L | Microbial resistance, dermatitis, endocrine disruption | Surface water bodies | [117] |
| Diclofenac, 17α-ethinylestradiol (EE2), 17β-estradiol (E2) | <1 ng/L | Biodiversity reduction of sensitive aquatic species | Human skin | [118] |
| Caffeine | 5–400 mg | Decrease in HepaRG cell viability after oral and dermal absorption. | Human skin | [119] |
| Chlorpyrifos (CPF) | 50 µM | Induces redox imbalance altering the antioxidant defense system in breast cancer cells | Rats | [120] |
| Ethylene-bis-Dithiocarbamate (Mancozeb) | 0–1000 ppm | Increase in (1) total malignant tumors, (2) malignant mammary tumors, (3) Zymbal gland and ear duct carcinomas, (4) hepatocarcinomas, (5) malignant tumors of the pancreas, (6) malignant tumors of the thyroid gland, (7) osteosarcomas of the bones of the head, and (8) hemolymphoreticular neoplasias; in Sprague-Dawley rats | Rat liver | [121] |
| BDE-154 (hexa-BDE) | 0.1–50 µM | Induces mitochondrial permeability transition and impairs mitochondrial bioenergetics in rat liver | Zebrafish | [122] |
| Tetrabromobisphenol A (TBBPA) | 0.4–1.0 mg/L | Induces developmental toxicity, oxidative stress, and apoptosis in embryos and zebrafish larvae (*Danio rerio*) | Zebrafish | [123] |
| Dodecyl dimethyl benzyl ammonium chloride (1227) & fatty alcohol polyoxyethylene ether (AEO) | 1 µg/mL | Toxic to locomotor activity on Zebrafish larvae | Water and Wastewater | [124] |

## 4. Analytical Methods of EPs

In conventional ways targeting at emerging contaminants, advanced ultra-sensitive instrumental techniques such as electrospray ionisation-mass spectrometry (ESI(NI)-MS), Liquid chromatography -tandem mass spectrometry (LC-MS/MS), PTV-GC-EI-MS, gas chromatography-nitrogen-phosphorus detection (GC-NPD) etc. are commonly practised but not used for monitoring periodically. The use of such analytical instruments offer higher possibilities for detection of multiple emerging pollutants, and improve the detection limits even with very low concentration [125,126]. Several sample preparation techniques are available for various emerging contaminant's extraction from water. However, SPE is one of methods which is well adapted for a wide range of analyte's analysis, having dissimilar physical, chemical properties, and polarities. Moreover, SPE has a variety of available sorbents, their higher capacity compared to liquid–liquid extraction (LLE) [127]. Table 4 shows some analytical methods used so far for the extraction of EP samples.

**Table 4.** Analytical methods used for quantitative analysis of emerging pollutants.

| Pollutant Type | Pollutant Name | Sources | Extraction Method | Analytical Instrument | Advantage | Recoveries | %RSD | Limit of Detection | Ref. |
|---|---|---|---|---|---|---|---|---|---|
| Pharmaceuticals | Ibuprofen | Tap water; river water | solid-phase extraction (SPE) | liquid chromatography–tandem mass spectrometry (LC–MS/MS); ESI(NI)-MS | | 22 | 5 | | [125,127,128] |
| | diclofenac | Raw wastewater | SPE | LC–MS/MS | - Can be applied to a wide range of biological molecules<br>- Fast scanning speeds | 84 | 7 | | [125,129] |
| | Salicylic acid | Treated water | SPE; SPE HLB | LC–MS/MS; ESI(NI)-MS | ESI(NI)-MS: can analyze large masses | 34 | 7 | | [125,127] |
| Antiseptics | Triclosan | Tap water | SPE | LC–MS/MS | | 82 | 4 | | [125] |
| Flame retardants | Triethyl phosphate | Treated water;Wastewater, surface and drinking water | SPE; Liquid liquid extraction | LC–MS/MS; PTV-GC-EI-MS | high analytical sensitivity | 92; 89–107 | 2 | 5–20 ng/L | [125,130,131] |
| | Tri-isobutyl phosphate | Tap water | SPE; | LC–MS/MS; LC-ESI-MS/MS | | 82; 20–103 | 7 | 0.3–4 ng/L | |
| | Tetraethyl ethylene diphosphonate | Raw water | SPE; SPME (Solid-phase microextraction) | LC–MS/MS; GC-NPD | enable detection at levels below the EU regulatory level of 0.1 $\mu g/L^{-1}$ | 70; 24–109 | 6 | 5–10 ng/L | |
| Personal care products | N,N-Diethyl-m-toluamide | Treated water | SPE | LC–MS/MS | | 70 | 4 | | [125] |
| EDC | Diehtylhexyl phosphate | Raw water | SPE | LC–MS/MS | | 58 | 6 | | [125] |
| | BPA | Seawater | SPE HLB | LC-QqQ-MS | | - | - | | [132] |
| | Estrone | wastewater | SPE HLB + LC-NH2 | GC-MS/LC-QqQ-MS | | - | - | | |

LC–MS/MS has several advantages; can be applied to a wide range of biological molecules and has fast scanning speeds. ESI(NI)-MS was used by several researcher since this analytical instrument can analyze large masses. Through PTV-GC-EI-MS, analytical sensitivity is greatly enhanced for analytes with low concentrations [128]. The efficiency

of extraction and the sensitivity and selectivity enable detection at levels below the EU regulatory level of 0.1 μg/L$^{-1}$ by GC-NPD and GC-MS/MS [131].

## 5. Treatment Methods for EPs Removal

The removal of EPs from the water and the potential formation of disinfection by-products determines the quality of drinking water supplies. EPs can be removed using various treatment methods, including biological, physico-chemical, and oxidation methods. Even so, most of the treatment methods have disadvantages, such as secondary pollution, high maintenance cost, and complicated procedures in the treatment [133]. Typical drinking WTPs (chlorination, filtration, and coagulation-flocculation) are less efficient at entirely removing EPs, such as PCPs, selected pharmaceuticals, and atrazine [97]. Research also discovered that metal salt coagulants (ferric sulfate and aluminium sulfate) were ineffective at removing compounds such as trimethoprim, sulfadimethoxine, and carbadox [134]. However, some studies demonstrated that treatment methods such as adsorption, nanofiltration (NF), and reverse osmosis (RO) on powdered activated carbon (PAC) and granular activated carbon (GAC) effectively removed the pollutants [81,87,88]. Recently, electrochemical oxidation (EO), along with other advanced oxidation processes (AOP), have been considered a promising technique for removing CEC from water and wastewater [135].

### 5.1. Biological Treatment

Since the early 1900s, microbial biomass used to degrade nutrients, contaminants, and organics in wastewater. Contrarily, the use of such biological treatment in drinking water is less common. Nonetheless, recent advancements are starting to expand the favorability, possibility, and applicability of biological treatment technologies of drinking water. These advancements include: (1) the high complexities and rising costs of managing water treatment for residuals, which is membrane concentrates; (2) the drive for a technique that effectively destroys contaminants rather than concentrating them (i.e., green technologies); (3) the introduction of new contaminants that are highly prone to biological degradation, such as perchlorate; (4) the emergence of membrane-based treatment systems that are highly vulnerable to biological fouling; and (5) regulations that limit DBPs formation [136–138].

Only polar contaminants discharged in the final effluent were removed during biological treatment [139]. It is well understood that the conventional activated sludge (CAS) is the most cost-effective method to degrade and eliminate contaminants. However, it does not eradicate micropollutants in sewage treatment [140,141]. In Europe, activated sludge with a 14-h hydraulic contact time could remove approximately 85% of estriol, 17β-estradiol, estrone, and mestranol [142]. Less than 10% of synthetic and natural estrogens were removed during the biodegradation process [143]. While some of the components were absorbed in the sludge, the majority were still soluble in the effluent. The use of CAS to treat pharmaceutical industry wastewater necessitated a lengthy hydraulic retention time [144,145]. Due to the limited operational requirements, the capital cost is cheaper when compared to advanced treatment. Even though CAS is less harmful to the environment compared to chlorination, it has some disadvantages, such as a tendency to produce higher amounts of sludge [146], high energy consumption, as well as possible formation of foam, color, and bulk sludge in secondary clarifiers [145].

Membrane bioreactors (MBRs) are another biological treatment efficient in eliminating bulk organics and could be used as hybrid systems or in conjunction with CAS. The primary advantages of MBRs over CAS are their capability to treat various wastewater compositions [147] and meet small footprint requirements [148]. MBR has been shown to achieve high biological oxygen demand (BOD) and chemical oxygen demand (COD) removal in pharmaceutical manufacturing facilities [149]. Research has found that membrane bioreactors could be used to remove estrogens from wastewater, moderate efficiency of 17β-estradiol was effectively removed up to 67% and estrone up to 91% [150]. Estrone-3-

glucuronide, 17-estradiol-glucuronides, and estrone-3-sulfate, on the other hand, were not significantly removed.

Enzyme-based treatment processes are used in a recent biological treatment that has the potential to remove EPs. Under optimal conditions, [151] discovered that laccase treatment in the presence of the natural redox mediator syringaldehyde completely removed diclofenac cytotoxicity. Furthermore, syringaldehyde and laccase were successfully converted the triclosan dichlorination product into 2-phenoxy phenol as a non-toxic polymer [152]. Laccase-poly (lactic-co-glycolic acid) nanofiber in the presence of syringaldehyde can be a viable method for diclofenac detoxification and removal from aqueous sources [153]. Recent research has discovered that a combination of cross-linked enzyme aggregate (combi-CLEA) associated with enzymatic cascade reaction could remove pharmaceutically active compounds (PhACs) with a high removal rate of more than 80% [154]. Since it is environmentally friendly, enzymes application for treating EPs has future potential.

*5.2. Physico-Chemical Treatment*

Physico-chemical processes are typically used in conventional water treatment to eliminate pathogens, control taste and odour issues, and reduce turbidity. These processes may also have the added benefit of lowering a load of micropollutants in finished drinking water. However, the removal is frequently insufficient [155]. Activated carbon, coagulation-flocculation, and membrane filtration are all part of the physico-chemical process. Coagulation-flocculation is a standard physico-chemical process required for water treatment. Current research on one of the emerging DBPs (halo-benzoquinone) revealed that coagulation could not wholly eliminate the precursor components from drinking water sources [156].

Meanwhile, a different study found that reducing halogenic acetic acids (HAAs) and THMs precursors by poly aluminum chloride under improved coagulation could only achieve the highest removal rates of 59% and 51%, respectively [157]. The adsorption process with activated carbon has emerged as a viable option for treating purifying drinking water and industrial wastewater [158]. The most frequently used adsorbent for treating biological and chemical pollutants in raw drinking water and industrial wastewaters are PAC or GAC. The application is favourable since their porosity, surface area, and chemistry are highly developed [159,160].

The adsorption process is prevalent and influential in water treatment since it is simple to design, produces no undesirable by-products, and is insensitive to toxic substances [161]. Although activated carbons are preferred to remove various EPs, their use is sometimes limited due to their high cost. Even after it has been depleted, activated carbon can be regenerated for future use. However, carbon will be lost during the regeneration process, and the resulting product might have a lower adsorption capacity compared to the freshly prepared activated carbon [162]. Natural water spiked with 30 pharmaceuticals and a bench-scale WTP simulation model were used to test 80 different EDCs. Except for oxidation via ozonation and chlorination and PAC, Leusch et al. [87] discovered no significant compound removal. Following that, Abd El-Gawad el al. [88] and Bolong et al. [163] confirmed that GAC regeneration or PAC dose is required to get a high removal rate.

RO and NF have received much attention in recent years. Membrane filtration technologies are now widely used to treat wastewater reclamation and drinking water as they effectively remove most inorganic and organic compounds [164,165]. The use of NF and RO membranes is a promising technology for removing EPs [85,166]. Nonetheless, the RO membrane is far more effective than the NF at removing EPs, but RO requires more energy during the process, making it less desirable.

Numerous studies have discovered that using RO and NF membranes in water treatment plants effectively removes EDCs and PCPs, sometimes by up to 95% [167,168]. Also, a different study demonstrated that the NF application in aqueous solutions could effectively remove pesticides (simazine, atrazine, and diuron) as well as atrazine metabolite (DEA) [169]. For all solutes, the NF membrane outperforms particle coagulation-

flocculation, dual media filtration, and sedimentation, with atrazine, has the highest retention rate, while diuron has the lowest [170]. The NF membrane was also used in another study to eliminate organic chlorine pesticides such as DDT from drinking water. The outcomes showed that DDT could be efficiently eliminated from water using an NF membrane, with up to a 95% removal rate [171].

Subsequently, Acero et al. [172] reported that regardless of the water matrix, the ultrafiltration (UF) membrane was the best for removing the selected herbicides. The NF and UF membranes proved to eliminate more than 90% BPA in drinking water with concentrations ranging from 60–600 µg/L. The mixture of humic acid and BPA hydrophobic adsorption mechanism was discovered to be effective for BPA retention [173,174]. Due to its size, it may be challenging to remove nanomaterials from water using conventional filtration without pre-coagulation [175]. However, it was determined that the UF membrane removed more than 99.6% of the $SiO_2$ nanoparticles [176]. The use of membrane filtration as the final process could improve the removal efficiencies for nanomaterials. Nonetheless, the membrane filtration process has the disadvantage of fouling, which causes flux reduction and increases operational costs [177].

*5.3. Oxidation Treatment*

Oxidation, which uses chemical oxidants such as ozone ($O_3$) and chlorine, is one of the fundamental techniques for removing EPs. The chemical reactions in water can be reactive, resulting in by-products. Therefore, before selecting this treatment, a careful selection of chemical oxidants is required. Because of its high oxidation capability, $O_3$ has been widely used in water treatment for color removal, disinfection, the degradation of many organic contaminants, and taste and odor control for drinking water. O3 will react with organic pollutants directly or indirectly with molecular $O_3$ and free radicals (hydroxyl radical OH) produced during $O_3$ decomposition, respectively [178].

A study conducted by de Jesus Gaffney et al. [179] discovered that an $O_3$ concentration of 2.5 mg/L could treat raw water at 20 different drinking WTPs across the States. In contrast, UV and chlorine concentrations of 40 mJ/cm$^2$ and 2.5 mg/L, respectively, were less effective. Chemical oxidation with $O_3$ is also efficiently treating various organic micropollutants in bench-, pilot- and full-scale drinking water and wastewater experiments [180–185].

Due to the enhanced generation of hydroxyl radicals and photon-initiated cleavage of carbon-halogen bonds, combining $O_3$-based advanced oxidation processes (AOPs) such as $O_3$/UV, photochemical, Fenton-type techniques, and $O_3/H_2O_2$ is more efficient than ozonation alone. Hence, AOPs are favored to treat recalcitrant compounds [184]. Ionizing radiation was considered as an attractive option among the available AOPs to degrade various toxic organic pollutants, such as nitrophenols [186,187], chlorophenols [188,189], and antibiotics [190,191] in aqueous solution, which draws concern in many countries.

The benefits of ionizing radiation technology include: (1) good penetration range in the water matrix; (2) having no additional chemicals; (3) being insensitive to color and suspended particles; and (4) the recalcitrant compounds may be degraded in situ by reactive species formed during water radiolysis [192,193]. The high price of radioisotopes, as well as safety concerns, are significant factors limiting their application. Two types of irradiators are commonly used, namely gamma (ɣ) sources (137Cs or 60Co) and electron beam accelerators (EB) [194]. The dose rate of EB is high, and ɣ rays are extremely penetrating. Moreover, the cost of energy for ɣirradiation is significantly higher [195]. As a result, EB is a viable candidate for practical application to alleviate public concerns about radioisotope safety measures [196].

UV photolysis (Suntest apparatus, Xe, and Hg lamps) has been successfully used to degrade organophosphorus pesticides, revealing mechanisms, diverse kinetics, and by-product formation. Although more toxic oxons were present in some cases [197], several pesticides were effectively degraded by low-pressure UV photolysis, such as pentachlorophenol, diuron, atrazine, clofenvinphos, and alachlor [198]. Nevertheless, pesticides in an aqueous solution could only be slightly degraded by a simple photolysis

process. These lighting sources outperformed the photolysis process alone when combined with $H_2O_2$ or Fe (III) [199].

TiO$_2$ is a semiconductor catalyst that has been extensively studied in heterogeneous photocatalytic processes, making it one of the advanced oxidation treatments' options (AOT). Chemical stability, non-toxicity, and low cost are all advantages of TiO$_2$. However, TiO$_2$ has the disadvantage of being in powder form, necessitating a separation stage after treatment in order to employ it as a photocatalytic material in wastewater purification by photocatalytic treatment. According to the findings of the study by Borges et al. [200], TiO$_2$ has strong photocatalytic activity for the removal of paracetamol from wastewater, with a removal rate of 99–100% after 4 h of irradiation.

### 5.4. Combine Treatment Processes

During water treatment at laboratory and pilot plant scales, numerous researchers have investigated some pharmaceutical drugs removal. Using a combination of treatment methods such as activated carbon adsorption, chlorine or O$_3$ oxidation, RO, and filtration, 90% of the antibiotics were removed [61,201–205]. Nevertheless, integrating specific treatment processes such as coagulation-flocculation with iron salts or aluminum and UV disinfection treatments did not achieve satisfactory contaminant removal levels [134]. Certain pharmaceuticals (ibuprofen, clofibric acid, and diclofenac) cannot be removed using flocculation with activated carbon adsorption alone [206]. Advanced oxidation processes using hydrogen peroxide (H$_2$O$_2$) and O$_3$ at 1.8 mg/L and 5 mg/L, respectively, achieved significant removals.

A study was conducted by Kovalova et al. [181] on a pilot plant scale in Germany to eliminate selected pharmaceuticals (diclofenac, clofibric acid, bezafibrate, and carbamazepine) from the source of drinking water. Either sand filtration in both anoxic and aerobic conditions or flocculation using iron (III) chloride (FeCl$_3$) were ineffective in removing the desired pharmaceuticals. On the contrary, it was discovered that ozonation is very selective in removing these polar compounds. The presence of psychoactive stimulatory drugs in raw and finished drinking water from a Spanish drinking WTP was assessed by Watanabe et al. [207]. They discovered that amphetamine-type stimulants (except MDMA (ecstasy)) were wholly eliminated during pre-chlorination, flocculation, and sand filtration steps, resulting in concentrations lower than their limits of detection (LODs).

Moreover, combined treatment methods of ozonation activated carbon adsorption and coagulation-flocculation (with FeCl$_3$) in removing EPs in pilot and drinking WTP was evaluated by Ternes et al. [180]. The EPs include bezafibrate, carbamazepine, diclofenac, and clofibric acid. The finding showed that the combined ozonation and activated carbon adsorption process successfully removed the pollutants.

A different study conducted by Luine et al. [208] on estrogen sorption and coagulation elimination performance processes using activated carbon was compared. Even in a hybrid system with NF membranes, they revealed that sorption by GAC and PAC was more effective compared to coagulation. A removal of 17α-ethinylestradiol (EE2) and 17β-estradiol (E2) in MBRs with and without PAC addition was investigated by Yang et al. [209]. The MBR alone removed EE2 and E2 at rates of 70.9% and 89.0%, respectively. With PAC, however, the EE2 and E2 rates of removal increased by 15.8% and 3.4%, respectively. The research also found that biodegradation was the most common method for removing EE2 and E2 in MBRs [209].

Subsequently, the oxidation of two pesticides (trifluralin and bromoxynil) in natural waters in a batch using O3 and O3 combined with H$_2$O$_2$ was investigated by Chelme-Ayala et al. [210]. The degradation levels for both pesticides were less than 50% based on the results. Nevertheless, the combined O$_3$/H$_2$O$_2$ process increased the degradation level. It was also discovered that adding a photocatalyst of titanium dioxide (TiO$_2$) to a coagulation-flocculation process increased the industrial chemical (1,4-dioxane) removal rate by two-fold in an hour. Moreover, at a UV dose of 0.35 WL$^{-1}$, a continuous flow reactor with a residence time of 39 min removed more than 60% of the 1,4-dioxane [211]. Pesticides, over

time, may pose a risk to human health via water and the atmosphere. Conventional portable water treatment methods, such as sedimentation, coagulation-flocculation, and dual media filtration, are less effective at removing pesticides residues [212]. By incorporating more advanced processes before pre-treatment, such as oxidation of $O_3$ or $H_2O_2$, membrane filtration, or granular activated carbon filtration, advanced water treatment could enhance the efficacy of typical water treatment [212–215].

Table 5 depicts the available water treatment processes for EPs as well as their removal performance. The CW had a removal efficiency of 42%, the AS had 62%, the RBC had 63%, and the WSP had 82%. Except for the WSP system, all of these technologies demonstrated seasonal variability in removing emerging contaminants. The WSP is the only system that is safe in both seasons, whereas all methods could potentially reduce the aquatic risk, according to an ecotoxicological assessment study [216]. During UV/$O_3$ treatment, a synergistic impact between $O_3$ and UV was detected in the selected trace antibiotics degradation process [177].

**Table 5.** Treatment processes for removing emerging pollutants.

| Types of EPs Compounds | Removal Treatment | Result | References |
|---|---|---|---|
| Pharmaceuticals, sunscreen compounds, fragrances, antiseptics, flame retardants, surfactants, pesticides and plasticizers | An extended aeration system (AS) and a rotating biological contactor (RBC), a constructed wetland (CW) and a waste stabilization pond (WSP) | The efficiency of removal was 42%, 62%, 63%, 82% for the CW, AS, RBC and WSP, respectively. | [216] |
| 4,4′-(Propane-2,2-diyl)diphenol, Nonylphenol, and 5-chloro-2-(2,4-dichlorophenoxy)phenol | Electrooxidation | Removal efficiency for selected emerging pollutants reached 73–89% | [221] |
| Pharmaceuticals (carbamazepine, flumequine, ibuprofen, ofloxacin, and sulfamethoxazole) | NF/ Solar photo-Fenton | Removal by NF produced a permeate containing less than 1.5% of the initial concentration of pharmaceuticals and application of solar photo-Fenton to this stream led to a reduction of 88% and 89% | [222] |
| Pharmaceutical (β-blockers) | $Fe^{2+}$/$O_3$ | β- blockers were completely degraded, when the removal rate of organic matter reached 30.6% and 49.1% for $O_3$ and $Fe^{2+}$/$O_3$, respectively. | [223] |
| Pharmaceutical (Antibiotics) | NF and UV/$O_3$ | High rejections of antibiotics (>98%) were obtained in all sets of NF experiments and UV/$O_3$ process achieved excellent removal efficiencies of antibiotics (>87%). | [177] |
| PFCs | MBR and PAC | Removal efficiencies of 77.4% for PFOS and 67.7% for PFOA were observed in PAC-MBR with PAC dosage of 30 mg/L. The increase of PAC dosage from 30 mg/L to 100 mg/L in PAC-MBR had increased the removal efficiency for PFOS or PFOA both to more than 90%. | [224] |
| Pharmaceutical (ketoprofen) | $O_3$/UV | $O_3$ highly contributed to the mineralization of small carboxylic acids. High (~90%) mineralization degree was achieved using the $O_3$/UV method. | [225] |
| Pharmaceutical (diclorofenac) | UF and photocatalytic ($TiO_2$/UV-A catalysis–) | Optimum diclorofenac removal at UV-A radiant power per unit volume 6.57 W/L, pH ~ 6 and $TiO_2$ loading near 0.5 g/L with maximum of diclorofenac molecular degradation and mineralization ~99.5% and ~69%, respectively. | [226] |

**Table 5.** *Cont.*

| Types of EPs Compounds | Removal Treatment | Result | References |
|---|---|---|---|
| Disinfection by-products (THMs) | UV/$H_2O_2$ | The degradation rates of 6 iodinated THMs in UV/$H_2O_2$ system were rather comparable and significantly higher than those achieved in the UV system without $H_2O_2$. | [219] |
| Pharmaceutical active compounds (PhACs) | NF | The overall rejection was approximately 31–39% and 55–61% for neutral carbamazepine (CBZ), and ionic diclofenac (DIC) and ibuprofen (IBU) respectively. | [227] |
| Pharmaceuticals | UV/$H_2O_2$ | Most of the compounds are degraded by 90% at UV doses between 500 (MP) and 1000 (LP) mJ/cm$^2$ and 10 mg/L hydrogen peroxide. | [228] |
| Perfluoroalkyl acids (PFAAs) | NF and GAC | Both virgin and fouled NF270 membranes demonstrated >93% removal for all PFAAs under all conditions tested. The F300 GAC had <20% breakthrough of all PFAAs in DI water for up to 125,000 bed volumes (BVs). | [229] |
| Pesticides | UV photolysis and NF | The combination of UV photolysis and NF allows the production of water with higher quality than the individual processes with global removals higher than 95% for all the spiked compounds throughout the treatment. | [230] |
| Pesticides (diazinon) | NH$_4$Cl-induced activated carbon (NAC) | Maximum adsorption rate was 97.5% of 20 mg/L diazinon adsorbed onto NAC at a low solution concentration of 0.3 g/L and short contact time of 30 min at neutral pH. | [218] |
| Industrial chemical (1,4-dioxane) | Coagulation-flocculation and photocatalysis | The addition of TiO$_2$ photocatalyst to a coagulation–flocculation water treatment process significantly increased 1,4-dioxane removal up to 100% within 1 h in a batch reactor and >60% of 1,4-dioxane was removed in a continuous flow reactor with a residence time of 39 min at a UV dose of 0.35 WL$^{-1}$. | [211] |
| Hormone (17a-Ethynyestradiol) | UV/$H_2O_2$ | The UV/$H_2O_2$ treatment was able to remove 90% of the 17a-Ethynyestradiol content within 30 min. | [231] |
| Pharmaceutical (Bezafibrate) | UV/$H_2O_2$ | The removal of bezafibrate is > 99.8% in 16 min under UV intensity of 61.4 μm cm$^{-2}$, at the $H_2O_2$ concentration of 0.1 mgL$^{-1}$, and neutral pH condition. | [232] |
| Hormones and pesticides | NF | High percent rejections (67.4–99.9%) were obtained for the pesticides and hormones, often independently of the water composition. | [233] |
| Hormones | NF and UV Photolysis | The use of NF in the treatment gives rejection at levels higher than 71% for all target hormones except estriol. Low pressure indirect photolysis with 100 mg/L of hydrogen peroxide was also efficient to degrade the selected hormones with percent degradations higher than 74% achieved for all the hormones, except nonylphenol (55%). | [234] |
| Pharmaceuticals and drug abuse | UF and RO | Iopromide (up to 17.2 ng/L), nicotine (13.7 ng/L), benzoylecgonine (1.9 ng/L), cotinine (3.6 ng/L), acetaminophen (15.6 ng/L), erythromycin (2.0 ng/L) and caffeine (6.0 ng/L) with elimination efficiencies >94%. | [61] |

<p align="center">**Table 5.** *Cont.*</p>

| Types of EPs Compounds | Removal Treatment | Result | References |
|---|---|---|---|
| Pharmaceuticals, hormones and BPA | GAC and UV | The removal efficiency:<br><br>(1) Carbamazepine = 71 to 93% using GAC and 75% using GAC followed by UV.<br>(2) Gemfibrozil = 44 and 55% using GAC and increased to 82% when GAC was followed by UV.<br><br>BPA = 80 to 99% using GAC or GAC followed by UV. | [66] |
| Pesticides | NF | The highest removal of diuron was achieved in the presence of intermediate ionic strength where an increase in diuron removal of 36.47% was obtained after the addition of 0.02 M of NaCl. | [217] |
| Pesticides | UF | The rejection coefficients for the four phenyl-urea herbicides were also determined, with values ranged from 50–90% for linuron to 10–50% for isoproturon, depending on the selected membrane and the operating conditions. | [171] |
| DBPs (Dichloroacetic Acid) | UV/$H_2O_2$/ Micro-Aeration | Removal efficiency greater than 95.1% of DCAA in 180 min under UV intensity of 1048.7 $\mu W/cm^2$, $H_2O_2$ dosage of 30 mg/L and micro-aeration flow rate of 2 L/min. | [220] |
| Caffeine, PPCPs and EDCs | Ozonation | Ozonation removed over 80% of caffeine, pharmaceuticals and endocrine disruptors. | [235] |
| BPA | UF | 75% removal using polysulfone-made UF membranes | [236] |
| EE2 | UF | 85% removal using polyvinylidene fluoride -made UF membranes | [237] |

UF, NF, and $NH_4Cl$-induced activated carbon (NAC) have been used to remove pesticides such as phenyl-urea [171], diuron [217], and diazinon [218]. The UV/$H_2O_2$ technique was investigated to remove 6 iodinated trihalomethanes (6 ITHMs), EE2, paroxetine, venlafaxine, pindolol, bezafibrate sotalol, and metformin. The 6 ITHMs degradation rates in the UV/$H_2O_2$ system were significantly faster than in the UV system without $H_2O_2$ [219]. Dichloroacetic acid (DCAA) decomposition in water using a UV/$H_2O_2$/micro-aeration process was also analyzed [220]. DCAA was unremovable by H2O2 oxidation, UV radiation, or micro-aeration. Still, an integrated technique of UV/$H_2O_2$/micro-aeration was practical and could completely degrade DCAA.

## 6. Conclusions

Current knowledge of the risks of EPs to the environment and human health has been carried out. However, there is still a gap in our understanding of EPs' long-term effects. EPs' fate and adverse impacts on human health and aquatic life are restricted and sparse, necessitating a greater study and knowledge. Due to the rapid development of EPs in the environment, implementing cost-effective and sustainable detection, risk assessment, and removal programs for all of these elements is difficult.

In the meantime, EPs have become a challenge in sustainable water management, where climate change and population growth exacerbate water sources issues. The primary concern with these contaminants is their micro size, as they are not effectively eliminated by conventional water treatment. WWTPs are critical in separating contaminants before they are discharged into the river system. Some, though, are very persistent in the water and hard to remove or biodegrade quickly. Pollutant residues will spread in the atmosphere and contaminate drinking water sources.

Therefore, future research initiatives should emphasis the pollutants that have the greatest impact on human health and the aquatic environment, allowing for integrated research to alleviate pollution inputs while optimizing available resources.

**Author Contributions:** Conceptualization, M.R.S., S.S., and M.S.M.; writing—original draft preparation, M.S.M., N.Z.A., and T.H.N.; writing—review and editing, N.Z.A., T.H.N., M.R.S., and S.S.; supervision, M.R.S., M.M., S.S., and A.A. All authors have read and agreed to the published version of the manuscript.

**Funding:** This work was supported by the Water Security and Sustainable Development Hub funded by the UK Research and Innovation's Global Challenges Research Fund (GCRF) [grant number: ES/S008179/1] and Universiti Teknologi Malaysia (UTM), grant number R.J130000.7609.4C241.

**Conflicts of Interest:** The authors declare no conflict of interest.

## Abbreviations

List of emerging pollutant abbreviations used throughout the manuscript:

| Abbreviation | Full form |
|---|---|
| 6 ITHM | 6 iodinated trihalomethanes |
| AOP | Advanced oxidation processes |
| AS | Aeration system |
| BOD | Biological oxygen demand |
| BPA | Bisphenol A |
| CAS | Conventional activated sludge |
| $CHCl_2Br$ | Bromodichloromethane |
| $CHCL_3$ | Chloroform |
| COD | Chemical oxygen demand |
| CW | Constructed wetlands |
| DBP | Disinfection by-product |
| DCAA | Dichloroacetic acid |
| DDT | Dichlorodiphenyltrichloroethane |
| DEA | Atrazine metabolite |
| DEET | Diethyltoluamide |
| DES | Diethylstilbestrol |
| DWTP | Drinking water treatment plants |
| E1 | Estrone |
| E2 | $17\beta$-estradiol |
| E3 | Estriol |
| EDC | Endocrine disrupting compound |
| EE2 | $17\alpha$-ethinylestradiol |
| EV | Estradiol valerat |
| GAC | Granular activated carbon |
| HAA | Halogenic acetic acid |
| $H_2O_2$ | Hydrogen peroxide |
| Hg | Mercury |
| LOD | Limits of detection |
| MBR | Membrane bioreactor |
| NAC | $NH_4Cl$- induced activated carbon |
| NF | Nanofiltration |
| PAC | Powdered activated carbon |
| PBDE | Polybrominated diphenyl ethers |
| PCB | Polychlorinated biphenyl |
| PCP | Personal care product |
| PFC | Perfluorinated chemical |
| PFOA | Perfluorooctanoic acid |
| RBC | Rotating biological contactor |
| RO | Reverse osmosis |

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
