# Peer review of "A Review on Emerging Pollutants in the Water Environment: Existences, Health Effects and Treatment Processes"

_water, doi:10.3390/w13223258_

Round 1
Reviewer 1 Report
1. The title of this paper is too broad, it does not cover all the environment. Moreover, it does not reflect the content properly. For example, it should include, sources, health effects, and treatment in the title. Most of the contaminants have been discussed from the perspective of water, so the word “Environment” is not suitable for water.
2. There is no point to include abbreviations in the introduction section.
3. Most of the contaminants in Table 2 are not emerging pollutants, they already exist and are discussed in many articles.
4. What criteria were followed by the authors to select emerging and new emerging pollutants?
5. The introduction section should provide more insights into emerging contaminants.
6. There are several latest reviews on emerging contaminants in the literature. The authors should discuss those reviews and provide a justification for this review in the introduction.
7. Please provide the detail how many years have been covered in this review and what keywords and search engine was used for the literature search.
7. Please provide a chart showing the number of publications per year for each kind of emerging pollutants? What were the trends?
8. In the section of sources, discuss each class of EPs under a separate heading.
9. In the section on toxic effects, discuss each class of EPs under a separate heading.
10. Regarding the table of toxicological studies, please include a column to mention what kind of assays were used for these studies?
11. Before the treatment section, add a section on analytical methodologies used for each class of emerging pollutants. Please discuss the methods used for the preparation of each class of pollutants, their advantages, and their limitations. Discuss each class under a separate heading. The table should include pollutant type, pollutant name, extraction method, analytical instrument, limit of detection, %RSDs, recoveries, actually detected concentrations, etc.
12. The authors should provide their critical insights throughout the manuscript.
In its current form, the manuscript requires substantial revision.
Reviewer 2 Report
This manuscript provides a thorough review on the emerging pollutants in the environment. The topic is really up-to date and deals with a compelling issue of the need to dispose of the huge amount of contaminants in the waters.
The manuscript is well organized and clear. The charts and tables are also informative and understandable. It summarizes well the current state on this research field and gives valuable contribution to the scientific literature.
My advise would be updating the references with even more recent papers in this area.
Reviewer 3 Report
Line 56. Table 1. List of emerging pollutant abbreviations used throughout the manuscript
CHCL3 |
Chloroform |
“Lapsus type” CHCl3 , "L" not capital letter
Hg, Mercury is not an Eps. See:
Minamata disease (M. d.) is methylmercury (MeHg) poisoning that occurred in humans who ingested fish and shellfish contaminated by MeHg in 1956 year.
- Bakir, S. F. Damluji, L. Amin-Zaki et al., “Methyl mercury Poisoning in Iraq,” Science, vol. 181, pp. 230–241, 1973.
- Harada, “Minamata disease: methylmercury poisoning in Japan caused by environmental pollution,” Critical Reviews in Toxicology, vol. 25, no. 1, pp. 1–24, 1995.
Line 61.- Table 2. Emerging pollutants groups and their major compounds. Cont.
Neccessary include also alquil-phtalates used as plasticizers.
Hill SS, Shaw BR, Wu AHB.The clinical effects of plasticizers, antioxidants, and other contaminants in medical polyvinyl chloride tubing during respiratory and non-respiratory exposure. Clinica Chimica Acta 2001; 304: 1-8 Hill SS, Shaw BR, Wu AHB.
Plasticizers, antioxidants, and other contaminants found in air delivered by PVC tubing used in respiratory therapy.Biomedical Chromatography 2003; 17: 250-262.
And Parabenes as antiseptics. See:
Safety assessment of esters of p-hydroxybenzoic acid (parabens).
Food and Chemical Toxicology Volume 43, Issue 7, July 2005, Pages 985-1015
Lines 224 and 251 Table 4.
BPA contamination has serious consequences for wildlife in general, not just humans.
BISPHENOL A EXPOSURE DURING EARLY DEVELOPMENT INDUCES SEX-SPECIFIC CHANGES IN ADULT ZEBRAFISH SOCIAL INTERACTIONS. J Toxicol Environ Health A. 2015; 78(1): 50–66. doi:10.1080/15287394.2015.958419.
Line 253.- 4. EPs Removal Treatment Methods.
First: the best solution to eliminate the problem of mercury contamination is not to use it.
In Line 317 and graphics (page 13):
Coupled removal techniques should also be included, such as Electrochemical

Reviewer 4 Report
Manuscript title: Emerging Pollutants (EPs) in the Environment – A Review
Manuscript Number: water-1417204
Journal Submitted: Water
Specific Comments:
Title:
How about revising the title? You need to take more insights from the last sentence of the abstract and revised your title more.
Abstract:
L 17-19: Meaningless sentence.
The abstract must summarize the leading techniques of removal, treatment strategies, potential consequences on human and ecological health. The present details in the abstract are too basic and do not add value. Please thoroughly revise it.
The keywords should also be revised.
Introduction:
L 34-40: So, you mean to say the term is applicable to aquatic Eps only?
L 44: Used? Or discussed?
L 45: NORMAN???
L 46: Please mention the name before citing a paper like this.
L 50-51: How come that a pollutant exists and is yet to be detected?
L 52-54: There is something wrong with the description here.
L 56: Table 1. Please cite the references for each EP that you have listed here.
Table 2. Please create a separate column where you can mention the main groups only and then mention their pollutants classified in each group.
L 66: Which research?
L 84-90: Please revise the aims and write in the future tense.
- Sources of EPs and their Occurrence in Water Resources
L 92-95: I cannot see any connection within this sentence nor its link with the heading.
You have totally missed the sources and stick to “occurrence”.
Figure 1. What is the source of this figure?
Table 3. Which country of Europe? Either discuss with respect to continents or countries.
Table 3. Are there no EPs in Malaysia?
L 253: Removal or treatment?
Figure 2. How is this flow of events determined? Any references?
Conclusions:
Please make is sure that the conclusions are revised and summarize the main outcomes from this review.
Also describe the potential gaps in research, ways forward, and potential limitations of this review.
Do you propose any new techniques in the treatment of Eps?
References:
Okay.
Round 2
Reviewer 1 Report
Dear Authors,
Some of my comments have not been adequately addressed. I have highlighted those comments again. They are very important to be considered before the publication of this manuscript.
- Previous comment: There is no point to include abbreviations in the introduction section.
Authors’ response: Table 1 on abbreviations has been removed from the manuscript
Reviewer’s new comment: Please keep the abbreviations but not in the introduction.
- Previous comment: Most of the contaminants in Table 2 are not emerging pollutants, they already exist and are discussed in many articles.
Authors’ response: It is important to note that the majority of emerging contaminants are not pollutants that are totally new or have just gained entry into the environment. Rather, most emerging contaminants are well-established pollutants with a newly demonstrated toxic effect or mode of action. Thus, the word 'emerging' refers not only to the contaminant itself but also to an emerging concern about the contaminant. As such, emerging contaminants are often referred to as 'chemicals of emerging concern' or 'contaminants of emerging concern.'
Reviewer’s new comment: Please add this information to the introduction.
- Previous comment: What criteria were followed by the authors to select emerging and new emerging pollutants?
Authors’ response: The selection of emerging and new emerging pollutants was based on the criteria below.
- not necessarily new compound
- compounds that might have been present in the environment in the past for many years, but it is only during the last years that their presence was detected, and their significance started to attract interest
- compounds that are known for a longer time but their potential negative impact on humans and the environment was only recently realized
Reviewer’s new comment: Please add this information to the introduction.
- Previous comment: There are several latest reviews on emerging contaminants in the literature. The authors should discuss those reviews and provide a justification for this review in the introduction.
Authors’ response: Following paragraph has been added in the introduction section.
‘Sutherland and Ralph (2019) presented an extensive review on the microalgal bioremediation potentials of emerging pollutants. They demonstrated that microalgae have ability to concentrate, filter, eliminate or biotransform a wide range of emerging pollutantants. Gogoi et al., (2018) investigated the fate and occurrence of these contaminants in wastewater treatment plants and in the environment. The writers indicated out that future research should concentrate on the improvement of risk-based screening framework and models. Rodriguez-Narvaez (2017) carefully reviewed some treatment technologies such as biological, Phase-change, advance oxidation process for removal of emerging pollutants from water. In addition, microplastics are classified as emerging pollutants, the interaction between pollutants and microplastics was evaluated by Abaroa-Pérez (2018).’
Reviewer’s new comment: There are many latest reviews on emerging pollutants in the environment that you can find by simple google search. Please include those and then highlight what is new in your review (this should be clearly mentioned in the manuscript while discussing its scope). Links to some latest articles are given below:
https://www.sciencedirect.com/science/article/pii/S0165993619305722
https://www.mdpi.com/2073-4441/13/2/181 (This is a very similar article, please clearly mention the differences of your article).
- Previous comment: Please provide the detail how many years have been covered in this review and what keywords and search engine was used for the literature search.
Authors’ response: Following paragraph has been added in the Introduction section.
‘The keywords were used as emerging pollutants’s sources, effects, treatment, removal and searched in science direct and Scopus publications only within the last ten-year period. In this paper, peer-reviewed scientific literature on EP published from 2011 through 2021 were reviewed with special regards to their occurrence, detection methods, fate in the environment, and ecological toxicity assessment. A total of 4000 documents (among research articles and review papers) collected from Scopus database were analyzed’.
Reviewer’s new comment: Please mention how the representative research articles were selected out of 4000.
- Previous comment: Please provide a chart showing the number of publications per year for each kind of emerging pollutants? What were the trends?
Authors’ response: This is not a bibliometric analysis. This review requires specific writing on bibliometric analysis of publication breakdowns for each emerging contaminant, period of highest scientific productivity on the subject, most productive country in terms of number of works published, journals with highest number of publications, etc.
Reviewer’s new comment: The comment is very simple, please show the research trends with respect to different classes of emerging pollutants for the last 10 years selected for this review. All other information is not required.
- Previous comment: Before the treatment section, add a section on analytical methodologies used for each class of emerging pollutants. Please discuss the methods used for the preparation of each class of pollutants, their advantages, and their limitations. Discuss each class under a separate heading. The table should include pollutant type, pollutant name, extraction method, analytical instrument, limit of detection, %RSDs, recoveries, actually detected concentrations, etc.
Authors’ response: Added in section 4
Reviewer’s new comment: A simple table without enough critical discussion is provided. There are some incorrect statements such as instruments simplifies sample preparation. Establish a separate heading on each class of pollutants just like other sections and then specify what sample preparation and instrumentations are used for extraction and analysis. What are the advantages and limitations of each kind of analytical methodology? This section needs some hard work from the authors, and it will add value to the manuscript.
Reviewer 4 Report
The authors have sufficiently revised the manuscript.
Author Response
Thank you for your comments and suggestions..!
Round 3
Reviewer 1 Report
The authors have tried to address my comments.